# MicroRNAs in Leukemias: A Clinically Annotated Compendium

**DOI:** 10.3390/ijms23073469

**Published:** 2022-03-23

**Authors:** Aleksander Turk, George A. Calin, Tanja Kunej

**Affiliations:** 1Department of Animal Science, Biotechnical Faculty, University of Ljubljana, 1230 Domžale, Slovenia; turk.aleksander@gmail.com; 2Department of Translational Molecular Pathology, Division of Pathology, MD Anderson Cancer Center, University of Texas, Houston, TX 77030, USA

**Keywords:** microRNA (miRNA), leukemia, interaction network, miRNA-target interaction (MTI)

## Abstract

Leukemias are a group of malignancies of the blood and bone marrow. Multiple types of leukemia are known, however reliable treatments have not been developed for most leukemia types. Furthermore, even relatively reliable treatments can result in relapses. MicroRNAs (miRNAs) are a class of short, noncoding RNAs responsible for epigenetic regulation of gene expression and have been proposed as a source of potential novel therapeutic targets for leukemias. In order to identify central miRNAs for leukemia, we conducted data synthesis using two databases: miRTarBase and DISNOR. A total of 137 unique miRNAs associated with 16 types of leukemia were retrieved from miRTarBase and 86 protein-coding genes associated with leukemia were retrieved from the DISNOR database. Based on these data, we formed a visual network of 248 miRNA-target interactions (MTI) between leukemia-associated genes and miRNAs associated with ≥4 leukemia types. We then manually reviewed the literature describing these 248 MTIs for interactions identified in leukemia studies. This manually curated data was then used to visualize a network of 64 MTIs identified in leukemia patients, cell lines and animal models. We also formed a visual network of miRNA-leukemia associations. Finally, we compiled leukemia clinical trials from the ClinicalTrials database. miRNAs with the highest number of MTIs were miR-125b-5p, miR-155-5p, miR-181a-5p and miR-19a-3p, while target genes with the highest number of MTIs were *TP53, BCL2, KIT, ATM, RUNX1* and *ABL1*. The analysis of 248 MTIs revealed a large, highly interconnected network. Additionally, a large MTI subnetwork was present in the network visualized from manually reviewed data. The interconnectedness of the MTI subnetwork suggests that certain miRNAs represent central disease molecules for multiple leukemia types. Additional studies on miRNAs, their target genes and associated biological pathways are required to elucidate the therapeutic potential of miRNAs in leukemia.

## 1. Introduction

Leukemias are a group of cancers affecting the blood and bone marrow. There are many types of leukemia, but they are commonly categorized into four main subtypes: acute lymphoblastic (ALL), acute myelogenous (AML), chronic lymphocytic (CLL) and chronic myelogenous leukemia (CML). The unregulated proliferation of hematopoietic stem cells in the bone marrow causes anemia, fatigue, bruising, bone pain, higher susceptibility to infections and other symptoms [1]. A study on cancer statistics spanning 183 countries estimated over 470 thousand new cases of leukemia for the year 2020 [2]. While our understanding of leukemia’s molecular and cellular mechanisms has expanded, chemotherapy-based treatment options for leukemia types have largely remained unchanged. Meanwhile, targeted therapy-based treatments have seen significant development. An example of this is the development of FLT3 inhibitors for the treatment of AML, such as midostaurin, sorafenib and quizartinib [3]. Due to the cytotoxic side-effects of chemotherapy-based treatments and urgent need for the development of novel treatments has arisen.

MicroRNAs (miRNAs) are a class of short, noncoding RNAs approximately 22 nucleotides in length. Their primary role is post-transcriptional gene expression regulation. This is achieved by a complex of miRNA and associated proteins, which facilitate RNA cleavage. miRNAs are predicted to be involved in the regulation of over 60% of mammalian protein-coding genes [4]. Through this regulatory function they play a role in multiple biological processes important for cancer development, such as cell-cycle regulation, inflammation, apoptosis, stress response, cell differentiation and metastasis [5]. Increases and decreases in miRNA levels have been associated with an array of different diseases in humans, ranging from autoimmune disorders to myocardial infarction and cancers [6,7]. The role of miRNAs has also been studied in cancers, including leukemia [8,9]. Recent research has identified the Let-7 family of miRNAs as a potential diagnostic and prognostic marker for several types of leukemia. The Let-7 family has also been suggested as a potential therapeutic target to complement traditional treatment methods [10,11]. Several miRNAs have been identified as oncogenes while others have been shown to act as tumor suppressors [12]. Furthermore, altered expression and genetic variations in the miRNA regulome have been associated with cancer [13]. A growing body of knowledge thus indicates that miRNAs are a valuable resource for both our understanding of leukemia’s cellular mechanisms and potential novel therapeutic targets.

miRNAs were first associated with leukemia in 2002, when it was discovered that miR-15 and miR-16 genes are frequently deleted or downregulated in CLL patients [14]. Since then, a multitude of studies have been published on the role of miRNAs in various leukemia types. Over the past 20 years, a large number of miRNAs have been associated with leukemias and they have been shown to play regulatory roles in multiple signaling pathways important for the development of the disease [15].

As some leukemia-associated miRNAs are shared between different leukemia types, the aim of the present study was: (1) to provide an overview of the role of miRNAs in leukemias and the identification of miRNAs associated with multiple leukemias; (2) to identify miRNAs associated with multiple leukemia types and analyze their miRNA-target interactions (MTIs) in an MTI network; (3) to provide a review of clinical trials conducted on the role of miRNAs in leukemia. This would serve to highlight the role of miRNA in leukemia and identify central molecules with potential for therapeutic target development.

## 2. Results

In the present study we visualized the associations of miRNAs with several types of leukemia as a graphical network. We also visualized an MTI network of leukemia-associated miRNAs and target genes. Furthermore, we manually reviewed MTIs for studies conducted on leukemia and visualized a curated leukemia MTI network. Finally, we reviewed clinical studies on leukemia with miRNA as a study parameter. The study workflow diagram is presented in Figure 1.

A total of 137 unique miRNAs associated with 16 types of leukemia were retrieved from miRTarBase, displayed graphically as a network in Figure 2. The network contains 278 edges, meaning that some miRNAs are associated with more than one leukemia type. Leukemia nodes with the highest number of edges are AML, chronic lymphocytic B-cell leukemia (CLBL), leukemia and precursor cell lymphocytic leukemia/lymphoma with 76, 49, 40 and 24 edges respectively.

In total, 82 miRNAs were associated with only one leukemia type while 55 miRNAs were associated with two or more types. Among these, miR-125b-1 had the highest number of associations, as it was associated with 9 leukemia types. miR-17 had the second highest number of associated leukemia types—7. miR-18a, miR-19a, miR-19b-1, miR-20a, miR-34a, miR-92a-1 and miR-181a-1 were each associated with 6 different types of leukemia. miR-125b-2, miR-155, miR-181a-2 and miR-223 were each associated with 5 different types of leukemia. Among the 137 leukemia-associated miRNAs, 19 were associated with ≥4 leukemia types.

In order to visualize interactions between leukemia-associated miRNAs and targets, 86 unique protein-coding genes associated with leukemia types were retrieved from the DISNOR database. A total of 127 unique MTIs between leukemia-associated miRNAs and target genes associated with ≥4 leukemia types were identified using miRTarBase. These 127 unique interactions were between 37 leukemia-associated miRNAs and 49 leukemia-associated target genes (Figure 3). A figure with edges color-coded based on experiment type (cell lines, animal models, patient studies, etc.) is available in Appendix A. miRNAs with the highest number of MTIs are miR-125b-5p, miR-155-5p, miR-181a-5p and miR-19a-3p with 11, 7, 6 and 5 MTIs respectively. Genes targeted by the highest number of miRNAs include *TP53* and *BCL2,* which are targeted by 15 and 13 miRNAs respectively. Other protein-coding genes with a high number of interactions include *KIT, FBXW7, ATM, RUNX1* and *ABL1* which are all targeted by 5 miRNAs. A table of MTIs is presented in Appendix A, which includes: miRTarBase ID, miRNA name, target gene name, target gene Entrez ID, method of validation, support type and study reference (PubMed ID).

MTI data acquired from miRTarBase and DISNOR were then manually reviewed for MTIs identified in leukemia studies. The manually curated data included 64 MTIs (55 unique entries) between 27 miRNAs and 43 target genes. MiRNAs with the highest number of unique MTIs were miR-125b-5p, miR-181a-5p, miR-29a-3p and miR-15a-5p with 8, 7, 6 and 4 MTIs respectively. Genes targeted by the highest number of unique miRNAs were *BCL2*, targeted by 7, and *MYB*, targeted by 4 miRNAs. This manually curated MTI data was then visualized as a network (Figure 4).

In total, 27 clinical trials conducted on leukemia types that included miRNAs as a measurement parameter were obtained from the ClinicalTrials database, presented in Table 1.

## 3. Discussion

In the present study we visualized an MTI network of leukemia-associated miRNAs and target genes using data from the miRTarBase and DISNOR databases. The network contained 248 MTIs (127 unique entries) between 49 leukemia-associated protein-coding genes and 37 miRNAs associated with ≥4 leukemia types. miRNAs with the highest number of unique MTIs were miR-125b-5p, miR-155-5p, miR-181a-5p and miR-19a-3p with 11, 7, 6 and 5 MTIs respectively. Genes targeted by the highest number of unique miRNAs included *TP53* and *BCL2,* which are targeted by 15 and 13 miRNAs respectively. Other target genes with a high number of interactions included *KIT, ATM, RUNX1* and *ABL1*, each of which was targeted by 5 miRNAs. We also conducted manual review of the source publications of 248 MTIs in order to extract details regarding experiment types, study design and leukemia type.

### 3.1. Classification of Phenotypes in miRTarBase and DISNOR

MiRTarBase and DISNOR are manually curated databases. Both miRTarBase and DISNOR include some miRNAs and protein-coding genes associated with nonspecific leukemia types, such as “leukemia” and “acute leukemia”. DISNOR includes data from DisGeNET, which is a database of genes and sequence variants associated with human diseases and is also manually curated. As these databases draw from a vast variety of resources it is possible that primary sources did not include data specifying a leukemia type, or were observing multiple disease types simultaneously. As this work is primarily concerned with the general role of miRNAs and target genes in leukemia, target genes and miRNAs associated with nonspecific leukemia types were included in the initial data set.

### 3.2. Highly Connected miRNAs and Their Roles in Leukemia

In the present study, miR-125b-5p had the highest number of edges in the leukemia-associated MTI network. miR-125b can function both as a tumor suppressor and oncogene. In some types of cancer, such as mammary tumors, it is heavily downregulated. Meanwhile, in colon cancer and hematopoietic tumors, miR-125b is upregulated and contributes to the prevention of apoptosis [16]. miR-125b dysregulation has also been reported AML and ALL, where it was upregulated [17]. The role of miR-125b in leukemias is further supported by animal models, where the transplantation of miR-125b overexpressing cells into mice caused the development of leukemias in 12 to 29 weeks post-transplantation [18]. The importance of miR-125b in leukemia may come from its role in the regulation of signaling pathways, through which it controls cell proliferation, apoptosis, differentiation and metabolism. These signaling pathways include the ErbB2, PI3K/Akt/Mtor, P53, NF-Κb and the Wnt pathways, all of which play a role in cancer development [17]. Due to its multiple associations with leukemia and its related signaling pathways, miR-125b and the pathways it regulates may prove to be valuable targets for leukemia therapeutic approaches.

miR-155-5p plays a role in hematopoiesis, cell differentiation and immunity and has been shown to have oncogenic and proinflammatory functions [19]. A study on in vitro psoriasis cells has shown that the downregulation of miR-155 inhibited cell proliferation, migration and promoted apoptosis through the PTEN pathway. Additionally, the expression of multiple apoptosis related factors, such as PTEN, AKT, p-AKT, Bax and BCL2 was found to be significantly altered by miR-155 downregulation [20]. Furthermore, miR-155 has been shown to interact with the NF-Κb pathway as well as the ErbB2 and P53 pathways [21,22,23]. miR-155-5p overexpression has also been proposed as a biomarker for poor prognosis in CLL [19]. Results of a study on a murine model of lymphoma targeting miR-155 suggest it as a possible therapeutic target for lymphoma/leukemia. Nanoparticles and liposomes were used as a delivery system and results show an in vivo delay in lymphoid tumor growth [24]. miR-155-5p could therefore hold potential for future therapeutic approach development and warrants further research.

miR-181a-5p and miR-19a-3p have both been associated with numerous types of leukemia (Figure 2). miR-181a has been shown to target *PTEN*, a gene responsible for apoptosis regulation [25]. It has also been shown to play a role in the regulation of the Wnt, STAT3 and RAS/MAPK pathways, which have been implicated in leukemia development [26,27,28]. miR-181a has been shown to increase cell sensitivity to chemotherapeutic drugs by targeting several antiapoptotic genes [29]. Furthermore, miR-181a and other members of the miR-181 family have also been proposed as a potential therapeutic target for CLL [30]. Meanwhile, miR-19a-3p has been shown to promote tumor metastasis and chemoresistance via the PTEN/Akt pathway [31]. It has also been shown to modulate the PIK3IP1-AKT pathway, affecting cell growth [32]. A study on AML cell lines showed that the miR-19a-3p/HIF-1α axis can be targeted by simvastatin, resulting in the regulation of cell invasion, migration, proliferation and apoptosis [33]. In mixed-lineage leukemia (MLL) cell lines, anti-miR-19a dramatically reduced the colony-forming abilities of MLL cells compared to controls [34]. Further research and studies into the cellular mechanisms regulated by miR-181a-5p and miR-19a-3p are required in order to elucidate their role in leukemias and their value as potential targets.

miR-29a-3p has been associated with multiple types of leukemia. miR-29a displays a differential expression profile in CLL and AML [35,36]. Polymorphisms in the miR-29b-1/miR-29a cluster have also been associated with AML, specifically core-binding factor AML (CBF-AML) [37]. While miR-29a was proposed as one of five potential CLL biomarkers and is upregulated during early CLL development, the predictive abilities of this model were modest [38]. Another leukemia-associated miRNA is miR-15a-5p. In CLL, miR-15a is downregulated in the majority of cases [39,40]. In myeloid cell lines, miR-15a-5p was shown to confer chemoresistance by inhibiting daunorubicin-induced autophagy [41]. miR-15a-5p has also been shown to negatively affect cell survival and metastasis in CML patients. This is achieved by the interactions between the miRNA and *CXCL10*, which encodes a chemokine receptor responsible for stimulation of monocytes, NK and T cells [42]. In murine models, the deletion of the DLEU2/miR-15a/161 cluster resulted in the mice developing CLL [43].

### 3.3. Target Genes and Their Roles in Leukemia

In this study, we identified several target genes with a high number of interactions within the MTI network. The protein-coding gene with the highest number of these interactions was *TP53*, which is targeted by 15 miRNAs in the network. TP53 is a gene encoding a transcription factor that regulates several cellular functions, including apoptosis, DNA repair, changes in metabolism and cell cycle arrest. Sequence variants of *TP53* are some of the most common variants that occur in human cancers [44]. Leukemia patients with *TP53* sequence variants are a distinct patient group, as the presence of these variants makes leukemia notoriously challenging to treat. In a discovery cohort, decitabine, a cytidine analog, has shown promise in treating patients with *TP53* sequence variant AML and myelodysplastic syndromes (MDS). APR-246 is another potential therapeutic agent and has two ongoing phase II clinical trials [45]. While promising, these treatments are aimed at patients with *TP53* sequence variants, limiting their scope. Dysfunctions in miRNA regulation of *TP53* can also contribute to cancer development and it has been shown that multiple miRNAs affect TP53 expression [46,47]. *TP53* sequence variants also cause the dysregulation of the TP53/miRNA axis resulting in oncogenic effects. The targeting of miRNAs affected by this dysregulation has been proposed as a potential therapeutic target [48,49].

*BCL2* encodes a mitochondrial membrane protein responsible for caspase-dependent blocking of apoptosis in some lymphohematopoietic and neural cells. As such, it has been studied as a therapeutic target, with some success—venetoclax is a BCL2 protein inhibitor used for relapsed/refractory CLL with 17p deletion. BCL2 overexpression has also been detected in AML patients, but did not affect prognosis [50]. Venetoclax has also been shown to induce mitochondrial apoptosis in AML cells in combination with azacitidine [51]. miRNAs have however been highlighted as a potential therapeutic approach to chemoresistant cancers, however, an efficient method of delivery remains a challenge [52].

KIT is a receptor tyrosine kinase and plays a role in cell survival, proliferation, maintenance, hematopoiesis and other functions. MiR-193b has been shown to regulate *KIT* and repress cell proliferation in AML [53]. ATM is part of the PI3/PI4 family of kinases and is part of the cellular DNA damage response. It also regulates multiple proteins, including TP53. ATM has also been shown to regulate transcription of multiple miRNAs, including miR-205-5p and miR-335 as well as others through its regulation of TP53. Due to this regulatory function and involvement in cancers, the ATM/miRNA axis has been proposed as a potential therapeutic approach for treatment of radioresistant tumors [54]. *RUNX1* encodes a transcription factor that plays a role in hematopoiesis. Multiple miRNAs regulate *RUNX1* expression, including the miR-17-92 cluster and miR-27. Thus RUNX1 seems to be a central hub for specific miRNA regulatory circuits [55]. *ABL1* is a protooncogene encoding a tyrosine protein kinase. Multiple miRNAs, including miR-96, have been suggested as having potential therapeutic benefits for specific leukemia types [56,57].

*MYB* encodes a transcription factor that plays a role in cell proliferation, differentiation, signaling, apoptosis and is a key regulator of hematopoiesis. *MYB* has been shown to be over-expressed in patients with T-cell acute lymphoblastic leukemia (T-ALL). Similarly, *MYB* rearrangements were also detected in T-ALL patients [58]. Distal regulatory elements have also been shown to regulate *MYB* in human myeloid leukemia cell lines [59]. Among others, *MYB* has also been implicated in leukemia through its role in the survival of pre-B-cell acute lymphoblastic leukemia (pre-B-ALL). The loss of *MYB* decreased the viability of these leukemic cells. It has also been demonstrated that *BCL2* is a target of *MYB* in pre-B-ALL cells [60].

### 3.4. Biological Pathways Involved in Leukemias

Some miRNAs are associated with various types of leukemia, as seen in Figure 2. In total, 55 of the 137 miRNAs acquired from miRTarBase were associated with two or more leukemia types. Studies examining leukemia types have discovered that several pathways affect multiple disease types, which include the Notch, Wnt and the Hedgehog signaling pathways [61]. There are other pathways associated with multiple leukemia types, another example being the MAPK signaling pathway [62,63]. It has also been shown that miRNAs can play important regulatory roles in signaling pathways, including the MAPK and Notch pathways [64]. Due to this involvement in leukemia-associated signalization miRNAs warrant further research as potential therapeutic targets for leukemias.

### 3.5. Leukemia-Associated miRNAs and Target Genes and Their Roles in Other Diseases

In the present analysis we included studies describing 248 MTIs between leukemia-associated miRNAs and target genes. Many studies were performed on leukemias (64 MTIs); however, a larger portion was conducted on nonleukemia cancer phenotypes, including: breast cancer, gastric cancer, hepatocellular carcinoma, osteosarcoma, non-small cell lung cancer and prostate cancer. Additionally, many studies were performed on noncancer phenotypes, such as Epstein-Barr virus (EBV) infection, Alzheimer’s disease and astrogliosis. 17 studies were conducted in association with EBV infection and its effects on gene expression regulation. Interestingly, EBV has been correlated with more unfavorable prognoses for AML and CLL [65,66]. A complete list of phenotypes identified during manual article review is available in Appendix A. While 184 of 248 MTIs were not yet analyzed in leukemias, they could nonetheless represent candidate leukemia MTIs for future studies, as the miRNAs and target genes were reported to be associated with leukemia in separate studies.

### 3.6. MiRNAs Act as Both Tumor Suppressors and Oncogenes in Leukemias

The role of miRNAs in leukemia has been studied in the previously published literature [12]. miRNAs are known to have both tumor-suppressive and oncogenic functions in leukemia [12]. miR-15, miR-16 and miR-127 are known to have tumor suppressive functions while oncogenic miRNAs include miR-155, miR-21, miR-125b and miR-223 [12]. During oncogenesis the expression of some tumor suppressive miRNAs is decreased, which is attributed to their role in the regulation of genes responsible for tumor suppression [67]. Meanwhile, oncogenic miRNAs experience increased expression during oncogenesis and contribute to it by repressing tumor suppressor genes [12]. Due to these functions in leukemogenesis, miRNAs have been proposed as potential therapeutic targets for leukemias [68]. Targeting miRNAs has been proposed as a potential therapeutic approach for B-cell chronic lymphocytic leukemia (B-CLL) [68]. In B-CLL, the antiapoptotic protein BCL2 is overexpressed, resulting in reduced apoptosis levels [68]. *BCL2* is targeted by miR-15a and miR-16-1, whose downregulation or deletion has been associated with B-CLL development [69]. Similarly, other miRNAs are involved in regulating B-CLL-associated genes, including miR-21, miR-29 and miR-155, which are either overexpressed or deleted during the evolution of malignant clones in B-CLL [68]. Targeting *BCL2* and other oncogenes, such as TCL1, through their regulatory miRNAs could be a potential strategy for treating patients with B-CLL [68]. miRNA-based therapeutic approaches have also been proposed for the treatment of AML and ALL [70,71]. While miRNAs as leukemia therapy remains promising, a major barrier for their use is the lack of specific and efficient delivery methods, especially to tissues such as bone marrow [72]. Therefore, additional research into delivery methods for miRNA-based therapies is necessary.

### 3.7. The Use of miRNAs in Clinical Practice

Several miRNAs have been studied for their diagnostic, prognostic or therapeutic potential [73]. Results of a meta-analysis of miR-155 and its role in overall survival of AML and CLL patients indicate that high miR-155 expression is associated with lower overall survival of patients, making it a promising prognostic biomarker [74]. Six miRNAs were identified to have potential diagnostic and prognostic uses for AML. miR-10a-5p, miR-93-5p, miR-129-5p, miR-155-5p, miR-181b-5p and miR-320d were identified as differentially expressed in the serum of AML patients compared to healthy controls. Furthermore, the expression levels of these six miRNAs could differentiate between AML patients and controls. Additionally, miR-181b-5p serum levels were significantly associated with the overall survival of patients, highlighting it as a potential prognostic biomarker [75]. miRNA expression levels may also serve as indicators of chemotherapy resistance [52]. Results of a study on CLL showed that 53% of patients with low miR-34a expression levels were resistant to fludarabine treatment, though the mechanism for this chemoresistance has not yet been elucidated [76]. In CML progenitors, miR-486 upregulation resulted in elevated imatinib resistance [77]. Conversely, miR-30e upregulation sensitized K562 cells to treatment with imatinib through the regulation of the BCR-ABL protein [78]. While miRNA expression levels can be used to determine prognoses or identify better treatment options, their use as biomarkers is not yet routine. Studies on the role of miRNAs in leukemias have been extensively conducted on cell lines and animal models, but despite promising results, further development is required before miRNAs can be used in a clinical setting [72].

### 3.8. Clinical Trials on the Topic of miRNAs in Leukemia

We performed a database search of clinical trials involving leukemia and miRNAs (Table 1). In total, 27 such studies were available from ClinicalTrials.gov and include several types of leukemia. Of these, 14 entries were treatment and expression studies, 11 entries were biomarker-profiling studies and 2 were expression studies. All studies specified miRNAs as a measurement parameter during the study course. miRNA concentrations from various samples were observed during the course of treatment, though this treatment was focused on other cell components. Thus, no clinical trial presented in this study specifically targeted miRNAs as part of its treatment. Based on studies on the role of miRNAs in various leukemia types and their multitude of roles in leukemia-associated pathways, miRNAs represent untapped potential in leukemia treatment. Clinical trials based on targeting miRNAs in leukemia could lead to discoveries of novel therapeutic approaches, especially for leukemia types with poor prognoses and limited treatment options.

### 3.9. Study Limitations

While the present study conducts data synthesis on several important leukemia-associated miRNAs and their target genes, the study has some limitations. While miRTarBase and DISNOR are curated and extensive databases, keeping a large database completely up to date is challenging. Thus, recently discovered leukemia-associated miRNAs and targets may not be included in our initial data set. Similarly, articles from which these databases draw data may not accurately categorize leukemia types, as is the case of classifications such as “leukemia” or “acute leukemia”. In the future, these disease classifications would require manual review and updates to standardized disease phenotypes, such as: Human Phenotype Ontology, International Classification of Diseases and Disease Ontology databases. Furthermore, in order to present an organized and intuitive visual MTI network, we limited the miRNAs to those associated with ≥4 leukemia types. However, complete data acquired from miRTarBase and DISNOR is available in Appendix A and can be used to form an exhaustive visual network.

### 3.10. Future Directions

The role of miRNAs in leukemia has been extensively studied, however much of their role in the disease remains unknown. Our analysis revealed a highly interconnected MTI network (Figure 3), as the majority of miRNAs and target genes were part of a large interaction subnetwork. This high level of interconnectedness suggests that these miRNAs may play an important role in leukemia. Additional studies focusing on specific leukemias and their MTI networks could yield valuable results for identifying central molecules and biological pathways involved in disease development. Investigating a large array of miRNAs and target genes for differential expression would provide a larger data set, which would be invaluable in identifying factors that contribute to disease etiology and progression. Further studies into miRNAs and target genes in specific leukemia types could open novel diagnostic, prognostic and therapeutic options for patients.

While studies using miRNAs as therapeutic agents have been conducted, they are hindered by a lack of a reliable delivery method. This is especially true for tissues that are more difficult to selectively target, such as bone marrow. Additionally, a limited number of clinical trials targeting miRNAs in leukemia have so far been conducted. Further research into effective delivery methods, along with potential miRNA-based therapies, could therefore lead to useful novel treatment options for patients.

## 4. Materials and Methods

Data on miRNAs associated with leukemia was retrieved from miRTarBase, a manually curated database of experimentally validated MTIs and disease-associated miRNAs [79]. In miRTarBase leukemias are categorized into 16 types: acute biphenotypic leukemia, acute lymphoblastic leukemia, leukemia, acute leukemia, chronic lymphoblastic leukemia, chronic lymphocytic B-cell leukemia, B-cell leukemia, chronic myelogenous BCR-ABL positive leukemia, myeloid leukemia, acute myeloid leukemia, chronic-phase myeloid leukemia, acute promyelocytic leukemia, adult T-cell leukemia/lymphoma, precursor B-cell lymphoblastic leukemia/lymphoma, precursor cell lymphoblastic leukemia/lymphoma, precursor T-cell lymphoblastic leukemia/lymphoma. All 16 types of leukemia and their associated miRNAs were included in the data pool, including nonspecific types such as “leukemia” and “acute leukemia”. Cytoscape was used for the purpose of visualizing networks in this study [80].

miRTarBase was accessed to search for all MTIs of leukemia-associated miRNAs. These MTIs were then cross-referenced for target genes that have previously been associated with leukemia types. Leukemia-associated protein-coding genes were retrieved from the DISNOR database—a manually annotated database of genes associated with diseases [81]. DISNOR categorized leukemia into 16 types: leukemia; acute megakaryocytic leukemia; acute M1 myeloid leukemia; acute promyelocytic leukemia; childhood acute lymphoblastic leukemia; chronic lymphocytic leukemia; chronic neutrophilic leukemia; hairy cell leukemia; juvenile myelomonocytic leukemia; mast-cell leukemia; acute myelocytic leukemia; chronic atypical BCR-ABL negative myeloid leukemia; chronic myelomonocytic leukemia; chronic myeloid leukemia; precursor cell lymphoblastic leukemia-lymphoma; precursor T-cell lymphoblastic leukemia-lymphoma.

We thus obtained a data set containing experimentally validated MTIs between leukemia-associated miRNAs and leukemia-associated target genes. We then visualized an interaction network of MTIs. The network includes MTIs for miRNAs that were associated with ≥4 leukemia types.

The MTIs’ references were then manually reviewed for studies conducted on leukemia animal models, cell lines or patients. During manual review we summarized the role, phenotype association, phenotype group and additional information regarding each MTI. Additionally, we reviewed articles in miRTarBase 8.0′s archived data cache for MTIs associated with leukemia in order to encompass a larger array of leukemia-associated MTIs. The manually curated MTI data was then visualized as an interaction network.

Additionally, we conducted a database overview of clinical trials. Studies included leukemia types as a disease identifier and included miRNAs as either a therapeutic target or a measurement parameter, either during treatment or observational studies. Studies were acquired from the ClinicalTrials.gov database, which is developed and maintained by U.S. National Institutes of Health, Department of Health and Human Services, through its National Library of Medicine.

## 5. Conclusions

In the present study we conducted data integration and manual review of experimentally validated leukemia-associated miRNA interactions with their targets and visualized the results as a network. We also conducted a synthesis of data on miRNAs and their associated leukemia types. Furthermore, we acquired data on clinical trials involving leukemia and miRNAs. The analysis revealed a large, interconnected subnetwork of interactions identified in leukemia studies. Some miRNAs in the subnetwork were associated with multiple leukemia types, identifying them as central disease molecules. The results suggest that some miRNAs are shared as contributing factors to multiple leukemia types. miRNAs and target genes identified in this study are also associated with several biological pathways, suggesting these pathways could also be therapeutic targets. Some highlighted miRNAs have in fact already been the subject of studies on their potential therapeutic applications, though this has proven challenging. Currently the primary challenge to the use of miRNAs for leukemia treatment is the delivery method, as some tissues, particularly bone marrow, are difficult to selectively target. Research into delivery methods for specific tissues are therefore necessary for development of novel targeted therapy approaches. In the future, highlighted miRNAs, protein-coding genes and their associated biological pathways may prove valuable targets for therapies.

## Figures and Tables

**Figure 1 ijms-23-03469-f001:**
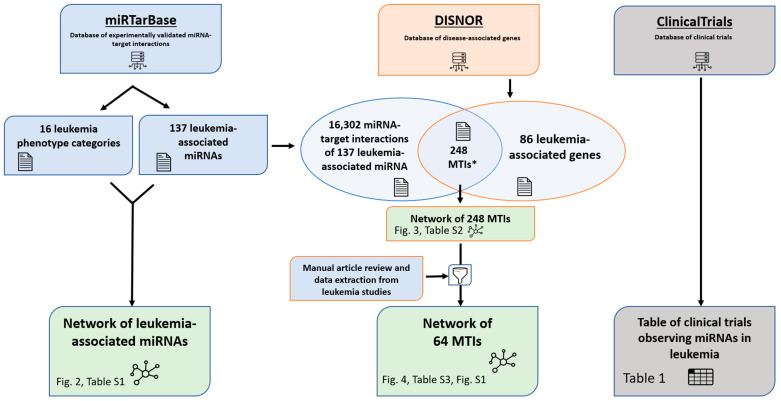
Study workflow diagram. * The 248 MTIs in between leukemia-associated miRNAs and target genes are limited to interactions with miRNAs associated with ≥4 leukemia types.

**Figure 2 ijms-23-03469-f002:**
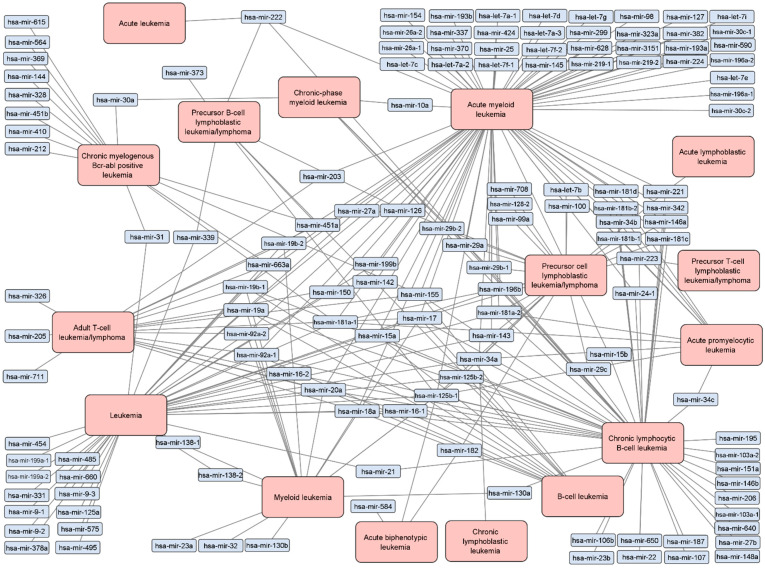
Network of experimentally validated associations between leukemia types and miRNAs. The network contains 137 unique miRNAs associated with 16 types of leukemia, as categorized in miRTarBase. Blue nodes are miRNAs while orange nodes are leukemia types. Each edge represents an experimentally validated association between the miRNA and corresponding leukemia type. A table of data is available in Appendix A.

**Figure 3 ijms-23-03469-f003:**
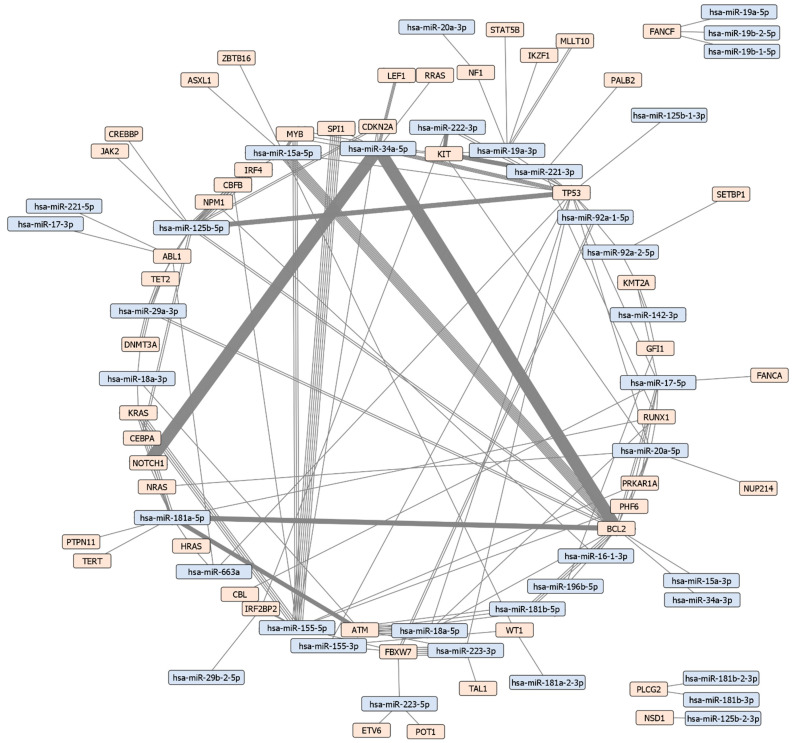
Leukemia miRNA-target interaction network. The network contains 127 unique MTIs between 37 leukemia-associated miRNAs and 49 leukemia-associated protein-coding genes. The total number of MTIs in the network is 248. Orange nodes represent miRNA while blue nodes represent target genes. Each edge is an experimentally validated interaction between the miRNA and its target, thus nodes with multiple connecting edges have been validated by multiple experiments.

**Figure 4 ijms-23-03469-f004:**
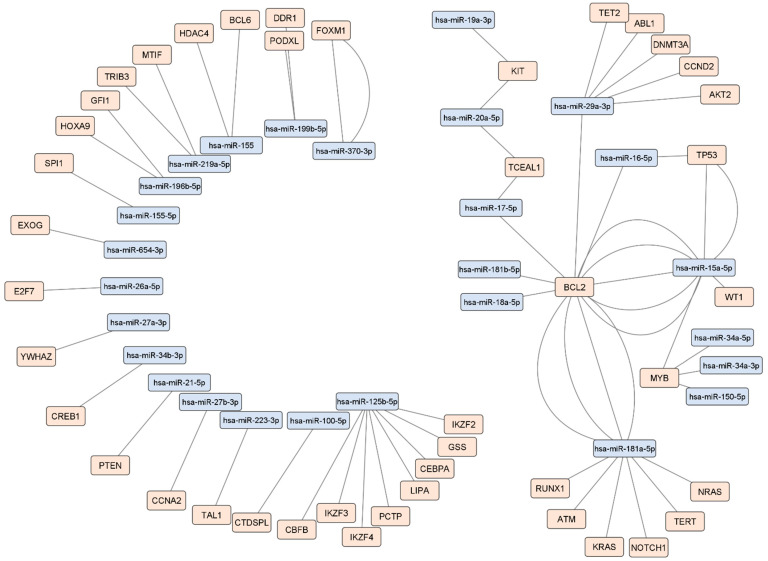
Network of manually reviewed MTIs identified in leukemia. The network is composed of 10 miRNA-target pairs, 5 sets of interactions between a single miRNA and two or more targets, and a subnetwork consisting of 12 miRNAs and 17 targets. Blue nodes represent miRNAs while orange nodes represent targets. Each edge represents a manually reviewed MTI identified in leukemia cell lines, animal models or patients. Edges do not differentiate between leukemia types. The leukemia type of each MTI is available in Appendix A.

**Table 1 ijms-23-03469-t001:** Clinical trials involving leukemia and miRNAs as study parameters. The table includes 27 clinical trials conducted on patients with varying leukemia types that simultaneously either targeted or monitored miRNAs throughout the trial course, data acquired from the ClinicalTrials database.

Study Start Year	Disease Type	Study Status	Study Approach	Therapeutic (If Any)	Age Group	Sample Source	Identifier
2010	Acute myeloid leukemia	Completed	Biomarker profiling		0–1	Tissue	NCT01229124
2012	B-cell acute lymphoblastic leukemia	Completed	Biomarker profiling		15–65	Tissue	NCT01505699
2010	Acute myeloid leukemia	Completed	Biomarker profiling		All	Tissue	NCT01057199
2016	Acute lymphoblastic leukemia	Unknown	Biomarker profiling		2–19	Blood	NCT03000335
2012	Acute myeloid leukemia	Completed	Biomarker profiling		0–3	Tissue, blood, bone marrow	NCT01511575
2011	Acute myeloid leukemia	Completed	Biomarker profiling		0–30	Bone marrow	NCT01298414
2007	Acute lymphoblastic leukemia	Unknown	Biomarker profiling		1–18	Not specified	NCT00526084
2010	Leukemia	Unknown	Biomarker profiling		3–21	Blood, cerebrospinal fluid	NCT01541800
2009	Acute myeloid leukemia	Active	Biomarker profiling		All	Blood, tissue	NCT00900224
2009	Acute myeloid leukemia	Active	Biomarker profiling		15–59	Blood, tissue	NCT00898092
2011	TEL/AML1-positive acute lymphoblastic leukemia	Completed	Expression study		1–18	Bone marrow	NCT01282593
2012	Acute myeloid leukemia	Completed	Treatment and expression study	Trebananib, low-dose cytarabine	18+	Blood, bone marrow	NCT01555268
2008	Acute myeloid leukemia	Completed	Treatment and expression study	Azacitidine, bortezomib	18+	Blood, bone marrow	NCT00624936
2008	Chronic myelomonocytic leukemia	Terminated	Treatment and expression study	Clofarabine	18+	Blood, bone marrow	NCT00708721
2016	Acute myeloid leukemia	Not yet recruiting	Treatment and expression study	Decitabine	65+	Blood, bone marrow	NCT02698124
2009	Acute lymphoblastic leukemia	Completed	Biomarker profiling		1–21	Blood, bone marrow	NCT00896766
2010	Acute myeloid leukemia	Completed	Treatment and expression study	Cytarabine, idarubicin, lenalidomide	18–64	Blood, bone marrow	NCT01132586
2014	Acute myeloid leukemia	Completed	Treatment and expression study	Pacritinib	18+	Blood, bone marrow	NCT02323607
2017	Chronic lymphocytic leukemia	Completed	Treatment and expression study	Cladribine, rituximab, vorinostat	18+	Blood, bone marrow	NCT00764517
2013	Chronic lymphocytic leukemia	Completed	Treatment and expression study	Bendamustine, rituximab	18+	Blood, bone marrow	NCT01832922
2009	Chronic myelomonocytic leukemia	Completed	Treatment and expression study	Azacitidine, lintuzumab	18+	Blood, bone marrow	NCT00997243
2012	Multiple hematologic malignanices	Completed	Treatment and expression study	Fludarabine phosphate, methoxyamine	18+	Blood, bone marrow	NCT01658319
2010	Multiple hematologic malignanices	Completed	Treatment and expression study	Azacitidine, bortezomib	18+	Blood	NCT01129180
2015	Acute myeloid leukemia	Active	Treatment and expression study		1–21	Blood, bone marrow	NCT02642965
2012	T-cell leukemia/lymphoma	Recruiting	Expression study		18+	Blood, bone marrow, skin tissue, tumor tissue, urine	NCT01676805
2018	Acute myeloid leukemia	Recruiting	Treatment and expression study	Daratumumab, donor lymphocytes	All	Blood, bone marrow	NCT03537599
2011	Multiple hematologic malignanices	Completed	Treatment and expression study	Histone deacetylase inhibitor 4SC-202	18+	Blood, bone marrow	NCT01344707

## Data Availability

Experimentally validated data on miRNAs associated with leukemia types was retrieved from miRTarBase, release 8.0 (https://mirtarbase.cuhk.edu.cn/) (accessed on 14 December 2021) [79]. DISNOR (release 2.0) was used to retrieve leukemia-associated genes (https://disnor.uniroma2.it/) (accessed on 19 December 2021) [81]. The visual representation of MTIs and the miRNA associations with leukemia types was made using the Cytoscape tool, release 3.9.0 (https://cytoscape.org) (accessed on 14 December 2021) [80]. Data on clinical trials involving leukemia and miRNAs was obtained from ClinicalTrials.gov (accessed on 15 January 2022). All the data presented can be found within the article and Appendix A.

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
