# Peer review of "MicroRNAs in Leukemias: A Clinically Annotated Compendium"

_ijms, 2022, doi:10.3390/ijms23073469_

Round 1
Reviewer 1 Report
it is an important work , not only bringing together and summarizing all available data on miRNAs in leukemia but more interestingly establishing links and perspective on their specificity and network interaction. This review is of great interest as a reference and analytical tool for all teams conducting research on the role of epigenetic regulation by miRnas in leukemias. The methodology used is rigorous and makes the work reliable and useful.
Author Response
We thank the reviewer for a positive response.
Reviewer 2 Report
In the manuscript „MicroRNAs in leukemias: a clinically annotated compendium“ authors conducted data synthesis using two databases with an idea to identify pivotal leukemia-associated miRNAs, as well as their target genes. They formed a visual network of 248 miRNA-target interactions (MTI) and manually researched the literature describing these for interactions identified in leukemia studies which was used to visualize 64 MTIs identified in leukemia patients, cell lines and animal models. Furthermore, they compiled leukemia clinical trials.
Broad comments: The authors report on an important matter, however the presentation of the data and the language make the review difficult to read.
Specific comments:
- Please improve the quality of figures. I am not able to read Fig2, Fig3 and Fig4 at all.
- The authors should avoid repetitions; “multiple types of leukemia” are mentioned throughout the text
- Many references are missing, eg. paragraph from lines 192 to 212 (line 192, 193, 197, 202, 204...)
- It would help if the authors would separate data on cell lines, animal models, and leukemia patients.
- Are there any data on KMT2A gene worth noting in the text?
- Discussion is difficult to read, maybe dividing it into subsections would help eg. miRNA and target genes should be separated.
- Conclusion: again repetitions “The miRNAs and target genes highlighted in this study could serve...“. and „miRNAs and target genes identified in this study are...“
Author Response
Reviewer 2:
Comment 1: “Please improve the quality of figures. I am not able to read Fig2, Fig3 and Fig4 at all.”
- Reply: Figures 2, 3 and 4 have been reconfigured to be more easily readable. Additionally, versions with improved resolution will also be uploaded to IJMS.
Comment 2: “The authors should avoid repetitions; “multiple types of leukemia” are mentioned throughout the text”
- Reply: We have reduced unnecessary repetitions. Improved language in lines 171, 221 and 242.
Comment 3: “Many references are missing, eg. paragraph from lines 192 to 212 (line 192, 193, 197, 202, 204...)”
- Reply: Additional references were added (lines 197, 205). A large part of this paragraph refers to articles by Yendamuri and Calin (2009) and Mardani et al. (2019), which extensively cover the role of miRNAs in leukemia. For improved clarity regarding references, more explicit referencing was added for specific sentences (lines 194, 195, 199, 201, 202, 203, 207 and 209).
Comment 4: “It would help if the authors would separate data on cell lines, animal models, and leukemia patients.”
- Reply: Data on the type of study conducted (cell lines, animal models, patients, tissue samples) is included in the Supplementary Data – Table S3. Additionally, we have color-coded the edges between nodes in a new network image, Supplementary Data – Figure S1.
Comment 5: “Are there any data on KMT2A gene worth noting in the text?”
- Reply: The focus of the present article is the general role of miRNA-target interactions in leukemias. Available studies on KMT2A (MLL) rearranged leukemias and their associated miRNAs are unfortunately however limited. Properly introducing MLL rearranged leukemias would require extensive additions to the paper, as they are a specific and complex topic. Consequently, the roles of miRNAs and KMT2A would be better suited for a future research article.
Comment 7: “Discussion is difficult to read, maybe dividing it into subsections would help eg. miRNA and target genes should be separated.”
- Reply: Thank you for the comment. We have added subheadings to the Discussion section. Subheadings now divide the Discussion on the topics of: disease classifications in databases, identified miRNAs and targets, the role of miRNAs in leukemia, study limitations and others. We have also expended the Discussion and added a new subsection for the potential clinical application of miRNAs for leukemia (treatment, diagnostics and prognosis).
Comment 8: “Conclusion: again repetitions “The miRNAs and target genes highlighted in this study could serve...“. and „miRNAs and target genes identified in this study are...“”
- Reply: Repetitions have been corrected. Adjustments were made to lines 455-456. We would like to thank you for your constructive comments.
Reviewer 3 Report
I have reviewed the manuscript. This is a nicely written paper, however, it needs to be improved.
1) I guess in the discussion and/or conclusions parts, the authors should more focus on the use of these MicroRNAs in clinical practice (e.g. diagnosis and prognosis).
2) The figures are in low resolution, they should be replaced with high quality ones.
Author Response
Comment 1: “I guess in the discussion and/or conclusions parts, the authors should more focus on the use of these MicroRNAs in clinical practice (e.g. diagnosis and prognosis).”
- Reply: We have expanded the Discussion section accordingly – a subsection has been added in order to discuss the potential use of miRNAs in clinical practice for prognostic, diagnostic and treatment applications. Additionally, the Discussion was divided into subsections.
Comment 2: “The figures are in low resolution, they should be replaced with high quality ones.”
- Reply: The figures have been reconfigured to be more easily readable (fig. 2, 3 and 4).